# Significant Variations in Double-Stranded RNA Levels in Cultured Skin Cells

**DOI:** 10.3390/cells13030226

**Published:** 2024-01-25

**Authors:** Shaymaa Sadeq, Suwalak Chitcharoen, Surar Al-Hashimi, Somruthai Rattanaburi, John Casement, Andreas Werner

**Affiliations:** 1Biosciences Institute, Newcastle University, Newcastle upon Tyne NE2 4HH, UK; s.k.sadeq2@newcastle.ac.uk (S.S.); s.o.t.al-hashimi2@newcastle.ac.uk (S.A.-H.); 2Fallujah College of Medicine, University of Fallujah, Al-Fallujah 31002, Iraq; 3Department of Microbiology, Faculty of Medicine, Khon Kaen University, Khon Kaen 40002, Thailand; suwalak.c@ku.th; 4Center of Excellence in Systems Microbiology, Faculty of Medicine, Chulalongkorn University, Bangkok 10330, Thailand; somruthai.r@hotmail.com; 5College of Medicine, University of Misan, Al-Sader Teaching Hospital, Amarah 62001, Iraq; 6Bioinformatics Support Unit, Medical School, Newcastle University, Newcastle upon Tyne NE2 4HH, UK; john.casement@newcastle.ac.uk

**Keywords:** double-stranded RNA, pattern recognition receptor, innate immunity, J2 antibody, RNA sequencing

## Abstract

Endogenous double-stranded RNA has emerged as a potent stimulator of innate immunity. Under physiological conditions, endogenous dsRNA is maintained in the cell nucleus or the mitochondria; however, if protective mechanisms are breached, it leaches into the cytoplasm and triggers immune signaling pathways. Ectopic activation of innate immune pathways is associated with various diseases and senescence and can trigger apoptosis. Hereby, the level of cytoplasmic dsRNA is crucial. We have enriched dsRNA from two melanoma cell lines and primary dermal fibroblasts, including a competing probe, and analyzed the dsRNA transcriptome using RNA sequencing. There was a striking difference in read counts between the cell lines and the primary cells, and the effect was confirmed by northern blotting and immunocytochemistry. Both mitochondria (10–20%) and nuclear transcription (80–90%) contributed significantly to the dsRNA transcriptome. The mitochondrial contribution was lower in the cancer cells compared to fibroblasts. The expression of different transposable element families was comparable, suggesting a general up-regulation of transposable element expression rather than stimulation of a specific sub-family. Sequencing of the input control revealed minor differences in dsRNA processing pathways with an upregulation of oligoadenylate synthase and RNP125 that negatively regulates the dsRNA sensors RIG1 and MDA5. Moreover, RT-qPCR, Western blotting, and immunocytochemistry confirmed the relatively minor adaptations to the hugely different dsRNA levels. As a consequence, these transformed cell lines are potentially less tolerant to interventions that increase the formation of endogenous dsRNA.

## 1. Introduction

The cycle of replication for most viruses involves double-stranded RNA (dsRNA) structures. Hence, the occurrence of cytoplasmic dsRNA is associated with viral infection and may trigger an innate immune response [1,2]. On the other hand, dsRNA is also produced from mitochondrial DNA and repetitive elements in nuclear DNA [2,3]. Mitochondrial DNA is transcribed in both directions, giving rise to complementary transcripts. Degradation of the noncoding sequences by SUV3 (mitochondrial ATP-dependent RNA helicase SUV3) and PNPase (Polynucleotide phosphorylase) reduces the levels of dsRNA and contains the remaining duplexes in the mitochondrial matrix. Functional impairment of SUV3 or PNPase due to mutations, drugs, or mitochondrial stress leads to an accumulation of dsRNA and leakage into the cytoplasm [3,4,5]. In the nucleus, dsRNA is predominantly formed from repetitive elements such as short and long interspersed elements (SINEs and LINEs) as well as endogenous retroviruses (ERVs) [2,6]. The transcripts from inverted repeats can fold back and form stem loops; alternatively, LINEs and ERVs can become bi-directionally transcribed and form hybrids in cis and -because of high copy numbers- in trans. Under physiological circumstances, transcription of the low complexity regions is minor with spurious dsRNA edited by ADAR1 p110 and contained in the nucleus. Increased transcription of repetitive elements in response to -for example- DNA de-methylation or a knockdown of ADAR1 leads to the export of dsRNA molecules into the cytoplasm [7].

DsRNA in the cytoplasm is recognized by various dsRNA-binding proteins that recognize structural differences between viral and endogenous transcripts to modulate a potential immune response [8]. The protein family of retinoic-acid-inducible gene-like receptors (RLRs) includes RIG1(retinoic-acid-inducible gene), MDA5 or IFIH1 (Interferon Induced with Helicase C Domain 1) and LGP-2/DHX58 (DExH-Box Helicase 58). Both RIG1 and MDA5 oligomerize upon dsRNA binding and signal via MAVS (Mitochondrial Antiviral Signaling Protein) to elicit an NF-κB (nuclear factor-kappa B)/IRF3/7 (interferon regulatory factors 3/7) mediated response. RIG1 recognizes short RNA hybrids with 5′ phosphate groups (up to ~300 bp), whereas MDA5 prefers long dsRNAs. Uncapped 5′ ends and extended perfectly complementary RNA duplexes are viral structural hallmarks; hence, RIG1/MDA5 predominantly recognize exogenous rather than endogenous dsRNA [2]. However, the discrimination can become blurred with increasing levels of cytosolic, endogenous dsRNA. LPG-2 displays regulatory effects rather than immune stimulation via interaction with dsRNA [9]. PKR (Protein kinase regulated by RNA, also known as eukaryotic translation initiation factor 2-alpha kinase 2, EIF2AK2), is an interferon-inducible protein that becomes activated by binding to dsRNA [10]. Binding of dsRNA induces dimerization of PKR and auto-phosphorylation. The activation leads to dissociation from the dsRNA and phosphorylation of the eukaryotic translation initiation factor 2 to inhibit global translation [11,12]. In addition, PKR activation leads to a type 1 interferon response via IRF3/7.

Endogenous dsRNA is implicated in various human diseases, potentially triggering an underlying inflammatory phenotype. For example, PKR is upregulated in Alzheimer’s disease [13], and its inhibition improves memory and cognition [14]. Conversely, PKR acts as a tumor suppressor and prevents oncogenesis via inhibition of translation [15]. Moreover, mutated RIG1 and MDA5 are associated with defects in innate immunity, confirming a tightly controlled balance between endogenous dsRNA formation and defense mechanisms [16].

Bidirectional transcription of endogenous retrovirus (ERV) stimulates endogenous dsRNA formation and interferon response, as shown in 63 cancer cell lines [17,18]. Since dsRNA-activated innate immunity may lead to apoptosis, hypomethylating drugs that promote dsRNA formation are widely used to treat hematological cancers and potentially other types of solid tissue tumors [19]. Similarly, induction of endogenous retrovirus transcription via HDAC inhibitors or demethylating drugs followed by dsRNA formation is harnessed to improve the action of checkpoint inhibitors [20].

A critical parameter that potentially shapes a dsRNA-mediated response to therapeutic or experimental interventions are endogenous levels of dsRNA and the related dsRNA sensor proteins [21]. Recent investigations into the dsRNA transcriptome of mouse liver and testis suggested significant organ-specific differences in both the amount and origin of dsRNA [22]. Given the importance of cell lines in establishing paradigms of cancer biology, we characterized the dsRNA transcriptome and the expression of sensor proteins in selected cellular models and found striking differences.

## 2. Materials and Methods

### 2.1. Cell Culture

A375 melanoma cells were kindly provided by M. Panagiotidis (Northumbria University, Newcastle, UK), and C8161 cells and fibroblasts were from PE Lovat (Newcastle University, Newcastle, UK). The cells were cultured in a DMEM high glucose medium (Sigma, Gillingham, UK) supplemented with 10% fetal calf serum (Sigma), 2 mM L-glutamine (Sigma), and 5 units/0.5 mg/mL penicillin/streptomycin (Sigma). Cells were grown at 37 °C in 5% CO_2_, and the medium was changed every 3 days. Passaging was performed when the cells reached 80–90% confluence. Cells were seeded in 6 well plates at a density of 250–500 × 10^3^ cells per well, in 96 well plates at 5–10 × 10^3^ cells, or in 50 mm dishes at 1 × 10^6^ cells per dish. For immunocytochemistry, cells were seeded on coverslips in 12 well plates at a density of 80 × 10^3^ cells per well.

### 2.2. RNA Extraction and RT-qPCR

RNA was extracted using the RNeasy kit (Qiagen, Manchester, UK) following the supplier’s instructions. The RNA was eluted from the columns with water and quantified using Nanodrop (Thermo Scientific, Paisley, UK). Samples were tested on 2% agarose gels in TAE buffer stained with Gel Green to ensure RNA integrity. For ᴄDNA synthesis, the Omniscript kit (Qiagen) with random hexamers (Qiagen) was used. Each 20 μL reaction contained 2 μg of total RNA in nuclease-free water up to 13 μL, 2 μL of dNTP (10 mM), 2 μL of 10× buffer, and 2 μL random hexamers (0.4 μg/mL) and 1 μL of Omniscript reverse transcriptase (200 U). Reactions were incubated at 37 °C for 1 h.

The expression profiling was performed by qPCR using a light cycler 480 (Roche, Burgess Hill, UK). Reactions were run in 96 well plates, each containing 0.5 μL of forward and reverse primers (10 μM), 1 μL of 1:5 diluted ᴄDNA, 3 μL of nuclease-free water, and 5 μL SYBR Green mix (Roche). The total reaction volume was 10 μL, and each condition was performed in three replicates. Actin was used as a reference gene. The cycling protocol was as follows: Activation of the polymerase at 95 °C for 10 min, followed by 45 cycles of 10 s at 95 °C, 20 s at 58 °C, and 5 s at 72 °C. Melting curve analysis was run to confirm the product. Primers were either designed using web-based tools or adapted from published studies and listed in Appendix A.

### 2.3. Western Blotting

Cells were grown in 50 mm Petri dishes followed by lysis in RIPA buffer (150 mM NaCl, 1.0% IGEPAL^®^ CA-630, 0.5% sodium deoxycholate, 0.1% SDS, 50 mM Tris, pH 8.0) and protein quantification was performed using a BCA protein assay (Pierce, Paisley, UK). 30 μg of protein lysate were run on CriterionXT precast 3–8% gradient gels (Bio-Rad, Watford, UK) with a protein ladder (Bio-Rad) in 1× tricine acetate running buffer (Bio-Rad). Proteins were transferred to PVDF membranes (Bio-Rad) using the Trans-Blot Turbo system (Bio-Rad). Membranes were washed in PBS-T (1× PBS, 0.1% Tween) and then blocked in 5% (*w*/*v*) skimmed milk in TBS-T for 1 h at room temperature. Primary antibodies against PKR (Abcam, ab32052, Cambridge, UK), RIG1 (Abcam, ab180675), MDA5 (Abcam, ab126630), ADAR (Atlas, B115763, Stockholm, Sweden) were used at a dilution of 1:5000 and anti-actin (Rabbit, Sigma; A2066) at 1:10,000. All antibodies were diluted in a blocking solution and incubated with the membranes at 4 °C overnight. After two washes in TBS-T, the secondary antibody (goat anti-rabbit IgG-HRP, 4030-05, 1:5000, Southern Biotech, Upper Heyford, UK ) in blocking buffer was applied at room temperature for 1 h. After washes in PBS-T, ECL (Bio-Rad) was added, followed by imaging using a chemiluminescence documentation system.

### 2.4. Immunocytochemistry

Cells were grown on coverslips (13 mm), washed with 1 mL PBS, and fixed for 10 min in 4% paraformaldehyde in PBS (Affymetrix, High Wycombe, UK). Cells were washed three times with PBS-T for 5 min, followed by permeabilization with 0.25% Triton-X-100 (Sigma) in PBS for 15 min. Samples were washed three times in PBS-T and blocked with ready-made 10% normal goat serum (Life Technologies, Paisley, UK) for 1 h at room temperature. Primary antibodies were diluted in blocking solution: J2 (monoclonal mouse IgG2a kappa chain; Scicons, Susteren, The Netherlands) and TOM20 (Abcam, ab186734) at 1:200 dilution, PKR (monoclonal IgG rabbit anti-human, Abcam, ab32052), MDA5 (monoclonal IgG rabbit anti-human, Abcam, ab126630) at 1:1000. After incubation at RT for 1 h cells were washed (three times 5 min in PBS-T) followed by secondary antibody incubation: For J2, an Alexa fluor™ 488 coupled goat anti-mouse IgG2a (Thermo) was used, otherwise an Alexa fluor™ 594 goat anti-rabbit IgG H + L (Thermo). Both secondary antibodies were diluted in blocking buffer 1:1000 and incubated for 1 hr at RT in the dark. After three washes, the cells were stained with DAPI (Vector Laboratories, Newark, CA, USA) and mounted on slides with ProlongTM Glass Antifade hard mounting medium (Thermo). Cells were imaged using an LSM800 Airyscan confocal microscope (Zeiss, Jena, Germany).

### 2.5. Spike in Probe

Plasmids encoding the natural sense–antisense transcript pair (*Slc34a2a* and *Slc34a2aas* from zebrafish) [23] were linearized with *Xba*I and transcribed in vitro using the MEGAscript T7 Transcription Kit (Invitrogen, Paisley, UK). The transcripts are 2607 bases (sense, NM_131624) and 1371 bases (antisense, NR_002876.2) long and share 563 bp of complementarity over three exons. The resulting RNA was quantified and mixed in equimolar concentrations to a total concentration of 0.4 μg/μL. One microliter was diluted 500× with 0.1 M NaCl, heated to 70 °C, and gradually cooled to hybridize the two strands. One microliter of the spike-in probe (0.8 ng) was added to the testis homogenate before J2 binding (see below).

### 2.6. Double-Stranded RNA Immunopurification

In essence, a published protocol was followed [3,22]. Cells were scraped and homogenized in 220 µL of NP-40 lysis buffer (50 mM Tris pH 7.5, 150 mM NaCl, 5 mM EDTA, 1% NP-40, 0.5% Na-deoxycholine, 220 units RNasin) using a syringe with 25G needle. 10% of the homogenate was used as input control, and total RNA was isolated using Qiazol according to the supplier’s protocol. The volume of the remaining sample was adjusted to 1 mL per sample with NET2 + DOC buffer (50 mM Tris pH 7.5, 150 mM NaCl, 1 mM MgCl2, 0.5% Na-deoxycholine, 0.2 units/µL DNase I). J2 antibody (5 µg, Scicons) and the spike in the probe were added. Samples were rotated for 3 h at 4 °C, and then 100 µL of µMACS Protein G MicroBeads (Miltenyi Biotec, Tokyo, Japan) were added and incubated for 1 h. µMACS columns were equilibrated with NP-40 buffer (50 mM Tris pH 7.5, 150 mM NaCl, 5 mM EDTA, 1% NP-40) followed by loading and washing the samples with 300 µL of NP-40 and 3 × 250 µL of wash buffer (50 mM Tris pH 7.5, 1 M NaCl, 1 mM EDTA, 1% NP-40, 0.5% Na-deoxycholine, and 0.1% SDS). DsRNA was eluted with hot water and purified using Qiazol (Qiagen). RNA was reverse transcribed using the SMARTer Stranded Total RNA-Seq Kit v3—Pico Input (Takara, Tokyo, Japan) and sequenced on a NextSeq 550 System (Illumina, San Diego, CA, USA).

### 2.7. Data Analysis

The quality of reads was assessed using FastQC, https://www.bioinformatics.babraham.ac.uk/projects/fastqc/ (accessed on 11 March 2022), and adaptors were trimmed using Trimmomatic (version 0.3.6) [24]. The spike probe reads in the different samples were aligned to a fragment of zebrafish Chromosome 1 (Chr 1: 14,432,434–14,454,662) that contains the *Slc34a2a* gene and the related natural antisense transcript (*Slc34a2aas*) originating from the bidirectional *Rbpja* promoter using STAR version 2.5.2b [25]. All data sets were then quantified using Salmon [26], and expression differences between the input samples of the cell lines were established using DESeq2 [27]. To establish the dsRNA transcriptome reads from all samples were mapped to the reference genome (GRCm38.p5) using STAR. Peaks and coverage were established with BEDTools genomecov (-bg -ibam) and multicov (BEDTools suite version 2.28.0; [28]). Regions with coverage greater than or equal to five times more than the background were annotated using ChIPpeakAnno (version 3.19.4) [29], Figure 1.

SQuIRE (Software for Quantifying Interspersed Repeat Expression, version 0.9.9.9a-beta) is designed to quantify transposable element (TE) expression at the subfamily and locus level [30]. The required tools (STAR 2.5.3a, BEDTools 2.25.0, SAMTools 1.1, stringtie 1.3.3, DESeq2 1.16.1, R 3.4.1 and Python 2.7) were installed using Anaconda 4.12.0. SQuIRE Fetch was used to download repeat masker annotation files, chromosome information files, and the STAR index using the following command (SQuIRE Fetch --build hg38 --fasta --rmsk --chrom_info --gene -v). This step was followed by aligning the RNA-seq data to the reference genome using SQuIRE Map (SQuIRE Map -1 R1.fastq.gz -2 R2.fastq.gz -r 141 -n file_name -b hg38, where r equals the maximum read length). SQuIRE Count was then used with the following options (SQuIRE Count -r 141 -n file.bam -s 1 -b hg38, where –s is the library format). Differential gene expression analysis was performed using SQuIRE Call (SQuIRE Call -1 treatment.file -2 control.file -A treatment -B Control -o output -i SQuIRE_count_folder). The fastq files have been deposited to the Sequence Read Archive (SRA), bioproject PRJNA1053654.

## 3. Results

We used the established protocol, including the antibody J2, to enrich dsRNA under non-denaturing conditions from the melanoma cell lines A375 and C8161, as well as from skin fibroblasts [3,22]. To control for enrichment, in vitro synthesized dsRNA with stretches of perfect complementarity, bulges, and single-stranded ends were included. The recovered RNA, including an input control, was reverse transcribed and sequenced. After quality control, between 21–39 × 10^6^ and 20–44 × 10^6^, reads were obtained for dsRNA enriched samples and input RNA, respectively, of which 50–70% mapped to the human genome or the zebrafish probe. The amount of the added probe was in the low ng range per sample, controlling for dsRNA immune enrichment with J2 and serving as a carrier for samples with low intrinsic dsRNA. For unknown reasons, the immune enrichment for one fibroblast sample failed and was excluded from the analysis. A summary of the read statistics is given in Appendix A.

The alignment of the spike in reads confirmed the validity of the enrichment protocol and also underpinned the observation that complementary regions are processed with reduced efficiency during library preparation and sequencing (Figure 2). The significantly smaller number of reads mapping to the antisense strand—one would expect sense-antisense to be roughly equimolar—may indicate that one antisense strand binds to several sense-strand molecules or reduced stability of the antisense transcript.

There was a striking discrepancy in read numbers between A375 cells, C8161 cells, and fibroblasts, with the cancerous cell lines expressing significantly higher levels of dsRNA than the fibroblasts. In fact, dsRNA levels in fibroblasts were so low that the spike probe accounted for almost all isolated dsRNA (Appendix A). The low abundance of dsRNA in fibroblasts was confirmed using dot blot analysis and underpinned by immunocytochemistry, where no J2 staining was found (Figure 3). Hence, fibroblast dsRNA reads were excluded from the peak-calling analysis (see below), where a certain read density is required to establish the background and generate a meaningful number of peaks.

Reads were then mapped to the human genome, and expression was quantified using a published peak-calling pipeline to determine the presence of dsRNA-forming transcripts (Appendix A) [22]. The dsRNA-associated reads mapped predominantly in promoter areas and in the 1000 bp downstream of genes. Fewer were mapped to exons, and less than 10% were each mapped to UTRs, introns, and intergenic regions. Hela cells [3] showed a different pattern, with most of the reads mapping to introns and intergenic regions (44.2% and 40.2%, respectively) followed by exons (7.1%) and less than 4% for the other biotypes (Appendix A, upper panel). All input samples (A375, C8161, and fibroblasts) showed a comparable distribution of biotypes with read mapping predominantly to gene flanking regions (promoters, immediate downstream) and exons, whereas 5′ UTR, 3′ UTR, introns, and intergenic regions were associated with <7% of the reads.

Endogenous dsRNA is essentially produced from mitochondrial transcripts and repetitive regions of the genome. To characterize the dsRNA transcriptome further, reads from both dsRNA-enriched and input samples were aligned to the human mitochondrial genome; in addition, the contribution of repetitive elements to the transcriptome was analyzed using SQuIRE [30]. In all samples, approximately 10–15% of total mapped reads align to the mitochondrial genome (Figure 4). This proportion increased slightly (median 15 vs. 10.5%) in the dsRNA samples, though the change was not significant. The quantification and comparison of mitochondrial coverage between samples is problematic since the total number of mitochondrial DNA copies is unknown. Likewise, the dsRNA-enriched samples lack robust internal standards for normalization; consequently, the results need to be interpreted qualitatively.

The second common source of dsRNA in cells is repetitive DNA, i.e., SINEs, ERVs, and LINEs. Reads derived from repetitive DNA elements, both single-mapping and multi-mapping reads, were characterized and quantified using the software package SQuIRE [30]. The pipeline establishes an expression profile using uniquely mapping reads and assigns the multimappers accordingly to the relevant repeat elements. An arbitrary cut-off at 100 FPKM was set, and both TE expression and TE subfamilies of Alu elements, ERVs, and LINEs were determined. Normalization of these data is challenging because the dsRNA data sets lack parameters that could be used for normalization; hence, total read numbers were used. The input data of the different cell populations did not show striking differences in relative expression levels of the TEs, though trends become apparent. First, C8161 cells show higher relative levels of Alus, SINEs, and LINEs than A375 cells and fibroblasts (Figure 5A). This is also reflected by greater subfamily variety (Figure 5B). Moreover, both relative expression and variety of TEs are slightly enriched in the dsRNA-enriched samples, except for LINE subfamilies in fibroblasts, which show the opposite trend (Figure 5). This outlier is likely the result of low read numbers in the fibroblast samples, especially affecting values around the cut-off. The findings so far suggest that in all tested cells, a comparable repertoire of repetitive elements is transcribed, though at a strikingly different level. This raises the question whether the variations in dsRNA levels shape a specific cellular background regarding dsRNA sensors and innate immunity.

Cells may be exposed to dsRNA in the form of viruses or from endogenous sources, including repetitive elements and mitochondria. To mitigate dsRNA in the cytoplasm, cells express proteins that recognize RNA hybrids to either reduce their immunogenicity or trigger a signaling cascade activating innate immunity (Figure 6). The expression levels of these dsRNA sensors were tested using RT-qPCR, Western blotting (for practical reasons only in A375 and C8161 cells), immunocytochemistry, and targeted pathway analysis.

Levels of PKR, MDA5, and RIG1 mRNA were slightly but still significantly lower in A375 than in C8161 cells, whereas for ADAR1, no difference was found (Figure 6A). At the protein level, only MDA5 showed significant changes, though C8161 cells displayed higher expression. The same trend, higher transcript, and protein levels in C8161 were also apparent for PKR and RIG1, though the changes in protein were not significant (Figure 6B).

A375 cells are larger than C8161 cells and divide at a lower rate. Immunocytochemical staining with both cell lines revealed a punctate cytoplasmic pattern of dsRNA-related fluorescence (Figure 7, red staining). The dsRNA does not obviously co-localize with the sensor proteins ADAR1 and MDA5 (signal overlap between 0.2 and 0.4). RIG1 in both cell lines and PKR in A375 cells, however, showed co-localization with dsRNA, predominantly in a ring around the nucleus (R^2^ of around 0.6 or greater). Of note, A375 cells showed significantly higher co-localization of dsRNA with RIG1/PKR than C8161 cells, in line with the higher levels of dsRNA in A375 vs. C8161 cells. Staining of fibroblasts with the J2 antibody revealed no signal despite clear detection of all sensor proteins assessed (ADAR1, MDA5, PKR, pPKR, and also TOMM20 to visualize mitochondria), in line with the very low read counts in the RNAseq experiments (Figure 3C).

Next, we compared the expression values of dsRNA sensors and related proteins in the RNAseq input samples and performed a KEGG pathway analysis focussing on RIG1-like receptor signaling. Interestingly, 2′, 5′ oligoadenylate synthase (2′, 5′ OAS) is significantly upregulated in both melanoma cell lines as compared to fibroblasts (Figure 8A). 2′, 5′ OAS binds to dsRNA and activates RNaseL, leading to cleavage of viral and cellular single-stranded RNA. Sustained activation induces autophagy and apoptosis. In addition, RNaseL represses the transposition of LINE1 in cell lines and reduces LINE1 RNA [31]. The ring finger protein RNP125, an E3 ubiquitin-protein ligase, is also significantly upregulated in the two melanoma cell lines as compared to fibroblasts (Figure 8B). RNP125 ubiquinates both RIG1 and MDA5, which leads to their degradation and blunts innate immune signaling [32].

To conclude, the striking differences in dsRNA levels between fibroblasts and cancerous melanoma cell lines influence the expression pattern of dsRNA sensor proteins and potential regulators. However, the changes are relatively subtle.

## 4. Discussion

We report the dsRNA transcriptome of two melanoma cell lines and compare it to primary dermal fibroblasts; published data from HeLa cells were analyzed in parallel as a benchmark. Our results suggest that the cancerous cell lines express significantly higher levels of endogenous dsRNA than the primary cells. This discrepancy led to minor changes in dsRNA sensor proteins that eventually shape or suppress a cellular innate immune response.

The impact of intrinsic dsRNA on innate immune signaling has been widely established in the context of various diseases, most prominently in the context of autoimmune disease and cancer and recently also in cellular senescence [33,34,35,36]. The focus of these reports is on dsRNA-associated proteins and downstream signaling rather than the dsRNA itself. However, the level and nature of endogenous dsRNA will shape the various protective mechanisms that feed into the activation of innate immunity and, as a consequence, how a cell reacts to changes in dsRNA due to disease or drug interventions [16].

Cancerous cells show increased expression of both LINEs and SINEs due to transcriptional activation and copy number amplification [37]. On the other hand, mitochondrial dysfunction -another hallmark of cancer cells- promotes the leakage of dsRNA into the cytoplasm and activation of the dsRNA sensor proteins PKR and MDA5 [3,38]. Increased levels of dsRNA in the two investigated melanoma cell lines (A375 and C8161) compared to dermal fibroblasts could, therefore, be expected, though the scale of the difference is still striking. At this point, we cannot predict whether different dsRNA levels are attributable to variations between normal and cancer cells or between cell lines and primary cultured cells; comparable data are lacking. Should our observation apply to other cancer versus un-transformed cells, alternative options to enhance the efficacy of immunotherapy may open. Rather than using adjuvants (Azacitidine, for example) to increase the already significant levels of dsRNA in cancer cells, pathways that clear dsRNA (ADAR, PNPase, or autophagy [39]) could be targeted [40]. The fact that non-cancerous cells express much less dsRNA may make them less susceptible to the drug with potentially milder side effects. Both Azacitidine and inhibition of ADAR1 (or knockdown under experimental conditions) lead to an increase in cytosolic dsRNA and activation of the OAS/RNaseL pathway [41].

Comparative expression and pathway analysis have established increased expression of OAS in the two melanoma cell lines, an unexpected finding in light of increased levels of dsRNA in these cells. However, the OAS/RNaseL pathway also plays a crucial role in cellular RNA turnover, and enhanced expression could be due to roles unrelated to dsRNA [42]. Accordingly, OAS is upregulated in a large number of primary tumors and enhances resistance against therapy-induced DNA damage [43]. In contrast, the ubiquitin ligase RNP125, which is also upregulated in the melanoma cell lines, triggers the degradation of MDA5 and RIG1, making the cells more tolerant towards a high dsRNA load (Figure 8B).

The dsRNA transcriptome has been investigated in various cell lines and whole organs, though most of these efforts focused on relative changes in physiological vs. pathophysiological conditions, for example [44,45,46,47]. The approach we applied here, including a competing dsRNA probe, produced an estimate of net levels of dsRNA in the three different cell samples. We did not count the cells before immune enrichment, though the striking difference in read numbers and the lack of dsRNA signals on dot blots and immunocytochemistry validate the reported discrepancy between cell lines and primary dermal fibroblasts. We have investigated the source of the dsRNA and confirmed the important contribution of mitochondrial transcripts and repetitive elements. The relative scale of mitochondrial dsRNA expression remains to be determined since the number of mitochondria and mitochondrial genome copies may vary in the different cell models. We also attempted to determine the proportion of light to heavy strands in response to oxidative stress by RT-qPCR as a proxy for dsRNA formation. Despite evidence that oxidative stress leads to the release of mitochondrial dsRNA and activation of PKR [39], we were unable to establish consistent changes in the proportion of light to heavy strand RNA in response to H_2_O_2_ (Appendix A).

Transcripts from repetitive elements were assessed using SQuIRE, and a trend of slight enrichment of TEs in the dsRNA samples was found. Transposon expression has been studied in detail in various cancers [48]. Their expression is greatly enhanced by histone deacetylases and DNA methyltransferase inhibitors, and many derive from unannotated, cryptic start sites [49,50]. Moreover, for acute myeloid leukemia, the ratio of TE [1] expression versus gene expression may have prognostic value for cancer progression. Of note, the low repeat cohort (low dsRNA) showed wide-ranging repression of immune response genes and poor prognosis. In contrast, the high-repeat patients (high dsRNA) showed reduced expression of tumor-related pathways and increased apoptosis [51]. Importantly, de-repression of transposons has been found in a wide range of diseases and is potentially responsible for the underlying chronic inflammatory phenotype [36,52,53,54]. TE expression and dsRNA also occur in pluripotent stem cells [55] and during hematopoietic development, though in the immune-privileged environment, other protective mechanisms appear to play a role, such as RNA methylation (m6A) [56].

The varied expression of dsRNA in cell lines did not cause strikingly different RNA and protein levels; the observed differences were mostly non-significant. These minor changes agree with the pathway analysis of differentially expressed genes between the three cell models (A375, C8161, and skin fibroblasts). None of the investigated dsRNA sensors was significantly changed; however, the ubiquitin ligase RNP125 that negatively regulates RIG1 and MDA5 was elevated in both transformed cell lines that have higher levels of dsRNA. Moreover, OAS is also upregulated to provide protection against high levels of dsRNA, which makes them highly responsive to increases in dsRNA, for example, induced by Azacitidine [41]. The finely balanced complexity of the dsRNA response is also documented by the significant expression changes of dsRNA sensors observed when cells are challenged with polyI:C or demethylating agents. The dsRNA stress caused significant expression changes depending on the nature of the cell lines and the particular stressors applied [57], thesis Dr Shaymaa Sadeq, Newcastle University). Evidence so far suggests that significant differences in dsRNA levels between individual tissues or specific cells within a tissue do exist [58] and may have physiological relevance. Though the nature of cells (healthy or pathological) or cell culture conditions could also have a determining effect on the dsRNA transcriptome, the underlying dsRNA sensors and the related immune signaling pathways, in essence, call for further research in the field.

Comprehensive characterization of the dsRNA transcriptome is hampered by significant technical difficulties. First, the immune enrichment depends essentially on lysis conditions. The breakdown of cellular compartments will mix and potentially hybridize RNAs that were not bona fide dsRNA molecules. Moreover, enzymatic processes, particularly reverse transcription, are negatively affected by dsRNA structures and may lead to a strand bias (Figure 2). Finally, bioinformatic analysis is challenging because of missing internal standards. For example, the quantification of mitochondrial reads is ambiguous because neither the number of mitochondria per cell nor the DNA molecules per mitochondria are usually known for a specific cell. Additionally, the quantification of nuclear reads is challenging if datasets from different biological systems were included. This may be achievable with a set of thoroughly tested spike-in probes added at the very start of the J2 enrichment procedure.

## 5. Conclusions

To conclude, we established a significant difference in dsRNA levels between two melanoma cell lines and primary dermal fibroblasts. However, the differences only lead to subtle changes in dsRNA sensing and signaling proteins. These findings may explain the discrepancies in susceptibility to dsRNA stress observed in different cell types. Our results highlight the importance of intrinsic dsRNA in shaping innate immunity and call for a comprehensive characterization of the dsRNA landscape in model systems such as cell lines, primary cells, or tissues. Moreover, experimental strategies that investigate the dsRNA transcriptome need to be standardized to allow for the comparison of different studies.

## Figures and Tables

**Figure 1 cells-13-00226-f001:**
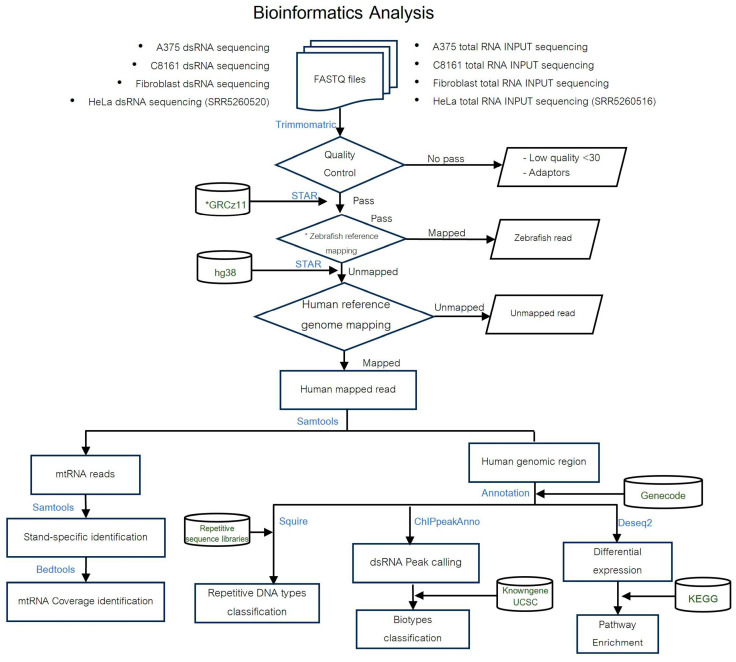
Bioinformatics analysis pipeline. The different samples are indicated at the top, and the software packages used are in blue. In addition to our own data sets from A375 cells, C8161 cells, and skin fibroblasts, previously published data from HeLa cells [3] was analyzed in parallel as an additional control. The zebrafish reference (*) encompasses bases 14,431,900–14,443,060 of chromosome 1 of the zebrafish genome assembly GRCz11.

**Figure 2 cells-13-00226-f002:**
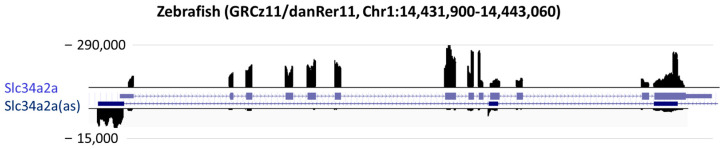
Reads mapping to the zebrafish *Slc34a2a* locus from the dsRNA spike in the probe. The scale for sense (*Slc34a2a*) is 290,000, and for the antisense (Slc34a2a(as)), it is 15,000.

**Figure 3 cells-13-00226-f003:**
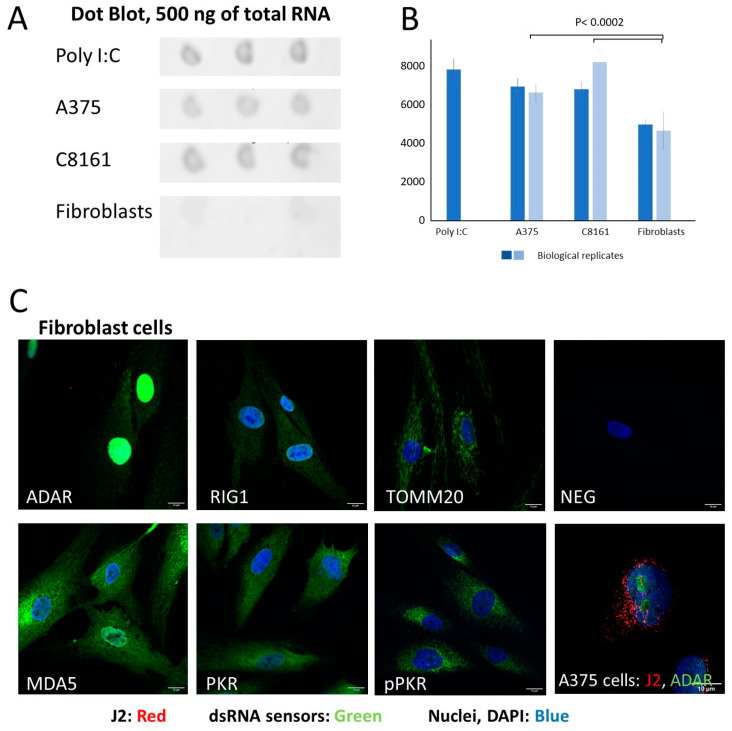
Fibroblasts express minimal amounts of dsRNA. DsRNA was assessed using the J2 antibody with dot blots and immunocytochemistry. (**A**) Three technical replicates of RNA were spotted on nitrocellulose probed with J2 and visualized using an HRP-coupled secondary antibody and CSPD star (Roche) detection. (**B**) Quantification of the luminescence signal in (**A**), three technical and two biological replicates (ANOVA with Tukey’s post hoc test). (**C**) Pictures 1–7, fibroblast cells; picture 8 (bottom right), A375 cells. The cells were fixed and stained with combinations of antibodies against dsRNA (J2, red) and dsRNA binding proteins (ADAR, RIG1, MDA5, PKR, and phosphoPKR) as well as TOMM20 (mitochondria) and DAPI (nucleus). In A375 control cells, the J2 signal is clearly visible throughout the cytoplasm, whereas it is undetectable in fibroblast.

**Figure 4 cells-13-00226-f004:**
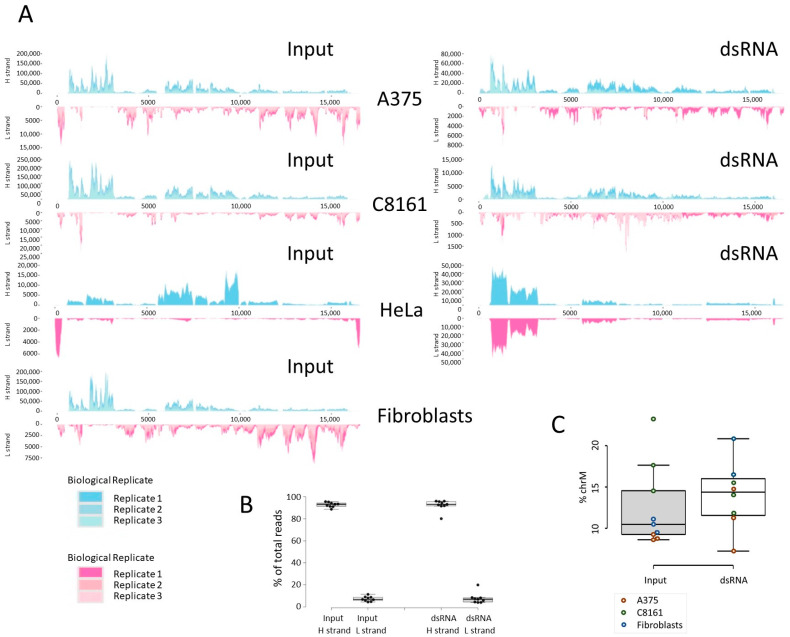
Summary of reads aligned to the mitochondrial genome. (**A**) Input and dsRNA reads of the three cell lines assessed, A375, C8161, and HeLa [3], as well as input from fibroblasts. (**B**) Total read numbers mapping to mitochondria, heavy (gene rich) versus light strand (gene poor). (**C**) Percentage of heavy (H) vs. light (L) strand mapping reads (*t*-test).

**Figure 5 cells-13-00226-f005:**
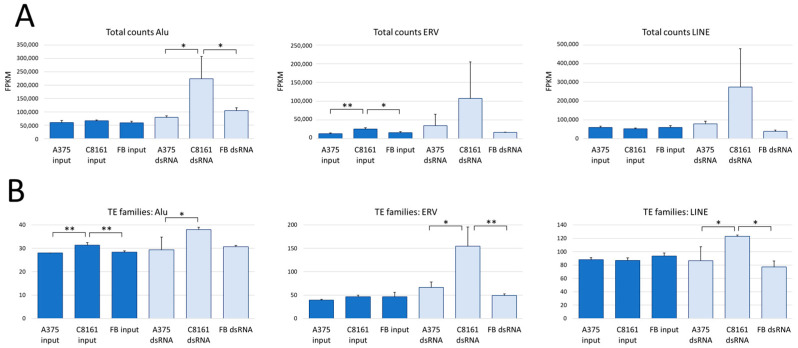
Expression of TE in A375, C8161, and fibroblast cells. (**A**) Total counts of normalized reads in FPKM mapping to Alu, ERV, and LINE elements. (**B**) Different TE families of Alu, ERV, and LINE elements in the three cell lines, A375, C8161, and fibroblasts. * *p* < 0.05; ** *p* < 0.01; ANOVA with Tukey’s post hoc test.

**Figure 6 cells-13-00226-f006:**
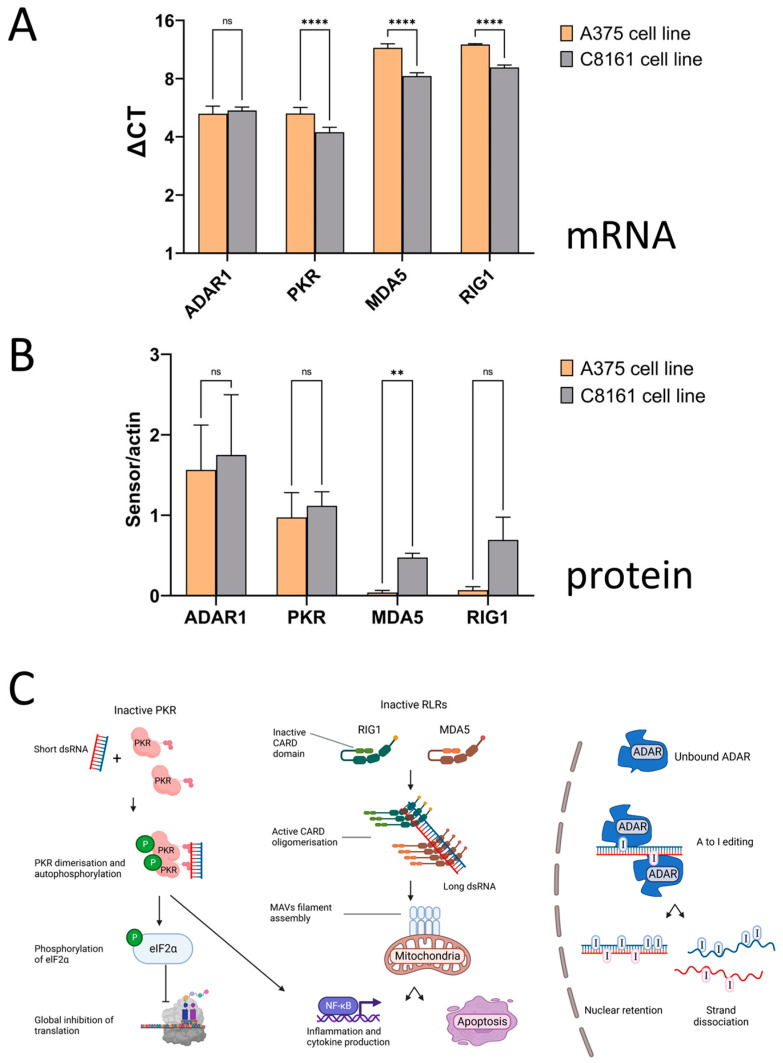
Double-stranded RNA sensors. (**A**) mRNA levels of dsRNA sensors in A375 and C8161 cells assessed using RT-qPCR. Actin was used as a reference. **** *p* < 0.001, ns, not significant, *t*-test. Of note, increased ΔCt values reflect lower mRNA expression levels. (**B**) Protein levels of dsRNA sensors assessed using Western blotting. ** *p* < 0.01, ns, not significant, *t*-test. Blots are provided in Appendix A. (**C**) Schematic diagram of dsRNA triggered pathways. Left, PKR recognizes short perfect RNA hybrids, dimerizes, and undergoes autophosphorylation. Phospho PKR induces global translational arrest via phosphorylation of eIF2α and leads to NF-κB mediated interferon signaling. Middle, the two Rig-like receptors (RLR) RIG1and MDA5 bind dsRNA with different specificity; RIG1 binds 5′ phosphate groups and polymerizes along dsRNA, whereas MDA5 binds long (500–1000bp) RNA hybrids. Both activate MAVS and lead to NF-κB mediated immune signaling and apoptosis. Right, dsRNA from nuclear DNA becomes edited by ADAR1 and is prevented from exiting the nucleus. Should the barrier be breached, MDA5 and PKR may become activated.

**Figure 7 cells-13-00226-f007:**
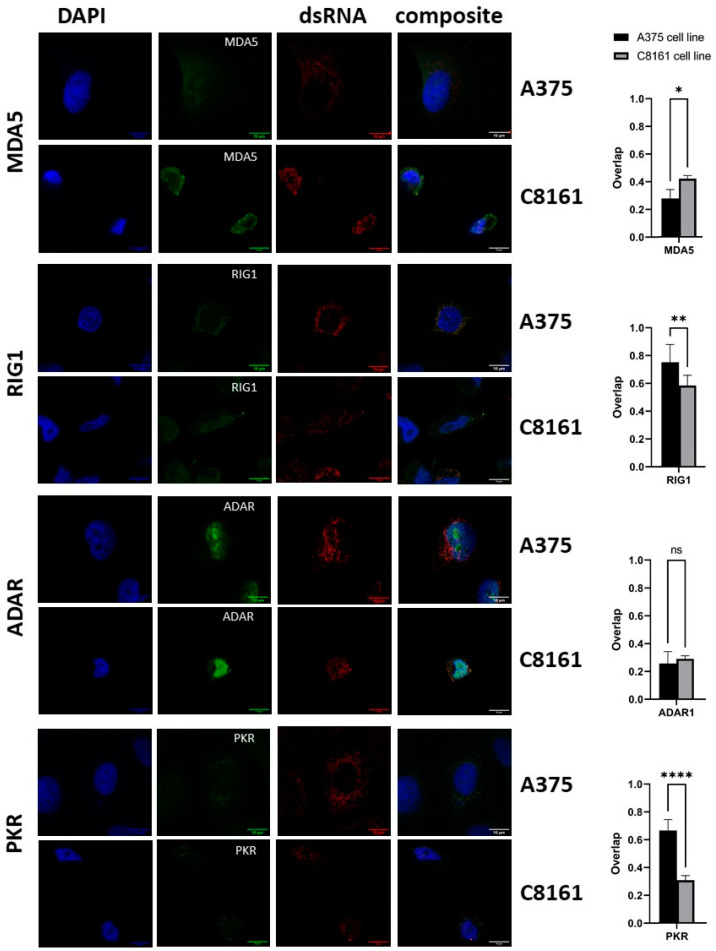
Immunohistochemical staining of dsRNA and dsRNA sensor proteins. The two cell lines, A375 and C8161, were assessed with the indicated antibodies against MDA5, RIG1, ADAR, PKR, and dsRNA (J2). The left panel shows nuclear DAPI staining (blue), followed by the specific dsRNA sensor proteins (green) and dsRNA (J2) in red. The panel on the right displays the composite. Scale bars are 10 µm (left panel). In the right panel, the spatial overlap of signals is compared between A375 and C8161 cells (R^2^). * *p* < 0.05, ** *p* < 0.01, **** *p* < 0.001, ns, not significant, *t*-test.

**Figure 8 cells-13-00226-f008:**
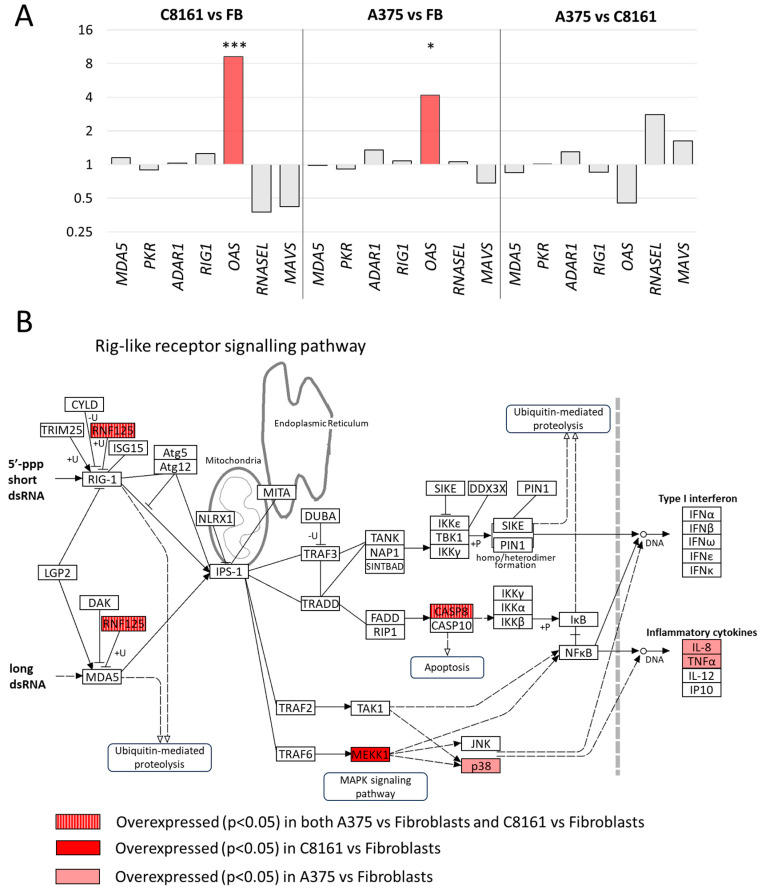
Expression differences in components of dsRNA signaling pathways between cell lines. (**A**) Expression differences as established using DESeq of selected dsRNA sensors, including MDA5, PKR, ADAR, RIG1, OAS, RNaseL, and MAVS. * adjusted *p*-value < 0.05; *** adjusted *p*-value < 0.005. (**B**) Components enriched in the KEGG pathway ‘Rig-like receptor signaling’ in C8161 cells versus fibroblasts and A375 versus fibroblasts. +P, phosphorylation; +U, ubiquitination; −U, deubiquitination. Adjusted *p*-value < 0.05.

## Data Availability

RNAseq data generated and discussed in this project have been submitted to SRA (accession number PRJNA1053654).

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
