# Peer review of "Significant Variations in Double-Stranded RNA Levels in Cultured Skin Cells"

_cells, 2024, doi:10.3390/cells13030226_

Round 1

Reviewer 1 Report

Comments and Suggestions for Authors

Authors conducted an enrichment and analysis of dsRNA in melanoma cells and primary dermal fibroblasts, uncovering notable variations in dsRNA levels between these cell types. The composition of dsRNA was notably influenced by both mitochondrial and nuclear transcription, with cancer cells displaying a lower contribution from mitochondria compared to fibroblasts. I found very interesting that despite significant differences in dsRNA levels between these cell lines, only slight discrepancies were noticed in the pathways responsible for dsRNA processing in the transformed cells. I agree that these findings imply a decreased tolerance in these cells towards interventions that increase the production of endogenous dsRNA and probably it can be a vulnerable point for therapeutic intervention.

Also the authors: The findings so far suggest that in all tested cells a comparable repertoire of repetitive elements is transcribed, though at a strikingly different level. This raises the question whether the variations in dsRNA levels shape a specific cellular background regarding dsRNA sensors and innate immunity.  I again find this encounter very interesting, because despite the inherent genetic instability of cancer cell it is very important to know that there might be differences that may greatly affect the results of the signalling process mediated by dsRNA, depending on the cell type. The context of dsRNA can influence the interpretation of dsRNA signals. 

This referee congratulates the authors for this nice and well-conducted work, I enjoyed reading the Introduction, and the exciting work.

Author Response

We would like to thank the reviewer for the kind comments and the excitement she or he shares with us about our findings.

Reviewer 2 Report

Comments and Suggestions for Authors

Dear authors,

1. In Abstract you use the abbreviations TE and OAS, however, the terms themselves are explained only on lines 206 and 364, correspondingly. I believe, abbreviations should be introduced right after the first mention of the term.

2. On line 63 the space after "RIG1" is absent.

3. The sentence on lines 64-67 seems to be incomplete or fused from the smaller ones.

4. HeLa cell line appears on Fig.1 without any previous explanation. What was the purpose of using it?

5. On Fig. 3C the scale bars are needed.

6. On line 284, I think, should be "reads", not "reds".

7. On lines 296 and 297 you repeat the explanations for SINEs etc, that have been already introduced.

8. The Fig.5 caption states * = p<0.05; ** p>0.01, however, no asterics could be seen on the figure. By the way, what is the purpose of indicating ** p>0.01?

9. On line 339 you are indicating the correlation of 35%. Do you mean R square? Anyway, 35% is too smal to be a considerable correlation.

10. On line 403 the dot is missing.

11. The Discussion seems to me too long and redundant.

Comments on the Quality of English Language

Some punctuation typos should me corrected.

Author Response

  1. In Abstract you use the abbreviations TE and OAS, however, the terms themselves are explained only on lines 206 and 364, correspondingly. I believe, abbreviations should be introduced right after the first mention of the term.

We have now spelled out TE (transposable element) and OAS (oligoadenylate synthase) in the abstract.

  1. On line 63 the space after "RIG1" is absent.

Done

  1. The sentence on lines 64-67 seems to be incomplete or fused from the smaller ones.

The sentence has now been rephrased into ‘Uncapped 5’ ends and extended perfectly complementary RNA duplexes are viral structural hallmarks, hence RIG1/MDA5 predominantly recognize exogenous rather than endogenous dsRNA.'

  1. HeLa cell line appears on Fig.1 without any previous explanation. What was the purpose of using it?

We have added a sentence to the legend explaining the use of HeLa cells: 'In addition to our own data sets from A375 cells, C8161 cells and skin fibroblasts, previously published data from HeLa cells [3] was analysed in parallel as an additional control.'   

  1. On Fig. 3C the scale bars are needed.

The figure has been amended.

  1. On line 284, I think, should be "reads", not "reds".

Done

  1. On lines 296 and 297 you repeat the explanations for SINEs etc, that have been already introduced.

This has been deleted.

  1. The Fig.5 caption states * = p<0.05; ** p>0.01, however, no asterics could be seen on the figure. By the way, what is the purpose of indicating ** p>0.01?

Thank you very much for spotting these errors. First, p>0.01 is a typo and should read p<0.01. Second, we analyzed the data in two different ways (one excluding potential outliers; however, this would have reduced statistical power) and eventually the wrong figure made it into the manuscript. The correct figure with asterisks has now been included.

  1. On line 339 you are indicating the correlation of 35%. Do you mean R square? Anyway, 35% is too small to be a considerable correlation.

The figure 7 has been updated with an analysis to determine special overlap of the signals (dsRNA with the relevant dsRNA-binding protein in A375 and C8161 cells) followed by comparison between the two cell lines. The following sentence has been added to the legend: ‘Right panel, spacial overlap of signals compared between A375 and C8161 cells (R2).’

The text has been changed to: ‘The dsRNA does not obviously co-localize with the sensor proteins ADAR1 and MDA5 (signal overlap between 0,2 and 0.4). RIG1 in both cell lines and PKR in A375 cells, however, showed co- localization with dsRNA, predominantly in a ring around the nucleus (R2 of around 0.6 or greater). Of note, A375 cells showed significantly higher co-localization of dsRNA with RIG1/PKR than C8161 cells, in line with the higher levels of of dsRNA in A375 vs. C8161 cells.’

  1. On line 403 the dot is missing.

A full stop has been introduced.

  1. The Discussion seems to me too long and redundant.

We have shortened the discussion and removed repetitive statements.

Comments on the Quality of English Language: Some punctuation typos should be corrected.

Done

Reviewer 3 Report

Comments and Suggestions for Authors

In the manuscript, the authors show that there are significant differences in dsRNA levels between the two melanoma cell lines (A375 and C8161 cells) and primary dermal fibroblasts. The experiments are almost well done and the results are reasonable. Therefore, I have only a few points that I think the authors should address. The specific points are as follows.

Major points.

1.      The intracellular dsRNA levels are indeed different between the two melanoma cell lines (A375 and C8161 cells) and the primary dermal fibroblasts. However, it is not known whether this difference is due to differences between cell lines and primary cultured cells, or between normal (non-cancerous) cells and cancer cells. The authors need to compare them with "normal cell lines". It is also better to use "primary" cancer cells. In any case, this concern could mislead the reader.

2.      The conclusion of the whole paper is somewhat unclear. It was understood that there were different variations between cells, but the abstract should include more quantitative results on the differences that were found.

Minor points.

1.      "Cultured cells" in the title of the manuscript should be changed to "cultured skin cells".

2.      Figure 6C: The Western blot data should be presented as the main figure.

3.      Figure 6: FBs have less intracellular dsRNA than other cells, but the authors need to show differences in how the amount of dsRNA sensors is affected in FBs as well.

4.      Figure 7: Overall, J2 antibody binding is weak. The authors need to increase the exposure to see clear dsRNA localization. Why is dsRNA only strongly stained when ADAR and dsRNA are co-stained?

5.      Abstract: "TE" should be spelled out.

Author Response

  1. The intracellular dsRNA levels are indeed different between the two melanoma cell lines (A375 and C8161 cells) and the primary dermal fibroblasts. However, it is not known whether this difference is due to differences between cell lines and primary cultured cells, or between normal (non-cancerous) cells and cancer cells. The authors need to compare them with "normal cell lines". It is also better to use "primary" cancer cells. In any case, this concern could mislead the reader.

This is a very valid comment, and we would wish to have a better understanding of cellular dsRNA levels and the associated machinery mitigating the consequences of endogenous dsRNA. To date, a search of GEO datasets identifies 34 independent studies, 12 in human and mouse each, 8 in C.elegans and 1 in rat and Drosphila, respectively, including the dataset from HeLa cells that was generated by A. Dhir in the Proudfoot lab. We used this dataset as a reference in our analysis. As indicated in the discussion, there are experimental and bioinformatic challenges associated with J2 enrichment and RNA-seq that makes a comparison of these independent datasets challenging and beyond the scope of this paper. We agree with the reviewer that either an in-depth comparison of all available dsRNA dataset with careful weighing of experimental factors and species differences, or a comprehensive study to establish a resource of comprehensive dsRNA transcriptomes. Such approach should include cell lines, primary cells as well as normal and cancerous tissue and a standardized experimental approach. Unfortunately, both of these approaches exceed the current capacity of this lab.

  1. The conclusion of the whole paper is somewhat unclear. It was understood that there were different variations between cells, but the abstract should include more quantitative results on the differences that were found.

Thank you for this comment. We have extended the Conclusion section to clarify the key points, i.e. (i) the significant differences in dsRNA levels and (ii) the need to generate comparable datasets and to establish standard analysis protocols. The conclusions read now: ‘To conclude, we established a significant difference in dsRNA levels between two melanoma cell lines and primary dermal fibroblasts. Though the differences only lead to subtle changes in dsRNA sensing and signaling proteins. These findings may explain the discrepancies in susceptibility to dsRNA stress observed in different cell types. Our results highlight the importance of intrinsic dsRNA in shaping innate immunity and call for a comprehensive characterization of the dsRNA landscape in model systems such as cell lines, primary cells or tissues. Moreover, experimental strategies that investigate the dsRNA transcriptome need to be standardized to allow for comparison of different studies.’ 

Minor points.

  1. "Cultured cells" in the title of the manuscript should be changed to "cultured skin cells".

Done

  1. Figure 6C: The Western blot data should be presented as the main figure.

Done, the order of the figures has been changed.

  1. Figure 6: FBs have less intracellular dsRNA than other cells, but the authors need to show differences in how the amount of dsRNA sensors is affected in FBs as well.

We agree with the reviewer that this is an important point. Unfortunately, we were unable to perform the suggested experiments due to limited availability of these cells. The fibroblasts are primary dermal cells from healthy skin of melanoma patients. The cells were a kind gift from Prof. Penny Lovat (Newcastle University) and supply was limited. In addition, these cells can only be used reliably up to passage 8 and we obtained them at passage 5. Hence we prioritized the J2 enrichment and RNA-seq experiments at the cost of RT-qPCR and Western blotting. To obtain an estimate of dsRNA sensors in fibroblast we extracted the values from the input RNAseq data and made the comparison for all three cells lines in figure 8A.

  1. Figure 7: Overall, J2 antibody binding is weak. The authors need to increase the exposure to see clear dsRNA localization. Why is dsRNA only strongly stained when ADAR and dsRNA are co-stained?

The figure has been changed to increase the red fluorescent signal (J2). This also sets the other figures in line with the ADAR/dsRNA staining.

  1. Abstract: "TE" should be spelled out.

Done

Round 2

Reviewer 3 Report

Comments and Suggestions for Authors

Major points.

>We agree with the reviewer that either an in-depth comparison of all available dsRNA dataset with careful weighing of experimental factors and species differences, or a comprehensive study to establish a resource of comprehensive dsRNA transcriptomes. Such approach should include cell lines, primary cells as well as normal and cancerous tissue and a standardized experimental approach.

I am not even asking for it. Please read my comment again.

I am asking whether a comparison of two melanoma cell lines (A375 and C8161 cells) with primary dermal fibroblasts is appropriate. I am not referring to the database nor a comprehensive study.

I am saying that it needs to be clarified whether the difference in this study is due to the difference between the cell lines and the primary cultured cells or the difference between normal (non-cancerous) cells and cancer cells.

Please answer me on this point. 

Minor points.

>Figure 6C: The Western blot data should be presented as the main figure.

>Done, the order of the figures has been changed.

Not done. I do not see any bands on the Western blot.

Author Response

Major points.

>We agree with the reviewer that either an in-depth comparison of all available dsRNA dataset with careful weighing of experimental factors and species differences, or a comprehensive study to establish a resource of comprehensive dsRNA transcriptomes. Such approach should include cell lines, primary cells as well as normal and cancerous tissue and a standardized experimental approach.

I am not even asking for it. Please read my comment again.

I am asking whether a comparison of two melanoma cell lines (A375 and C8161 cells) with primary dermal fibroblasts is appropriate. I am not referring to the database nor a comprehensive study.

I am saying that it needs to be clarified whether the difference in this study is due to the difference between the cell lines and the primary cultured cells or the difference between normal (non-cancerous) cells and cancer cells.

Please answer me on this point. 

We would like to apologize for the confusion. The reviewer has a very valid point to ask whether a comparison between our three model systems (A375 cells, C8161 cells and primary dermal fibroblasts) is appropriate. The short answer is ‘we don’t know whether cell culture differences or cancerous/non-cancerous origin are the reason for the divergent dsRNA transcriptomes’, there are not enough comparable data. However, we have detected significant differences in dsRNA levels in pathological vs healthy adjacent tissue in both melanoma and psoriasis (these manuscripts are in preparation and data will be available upon publication of two Newcastle University thesis from S.K. Sadeq and S. Al-Hashimi, respectively). We have also added the following sentences to the discussion to clarify the issue:

‘Evidence so far suggests that significant differences of dsRNA levels between individual tissues or cells do exist. Though, healthy or pathological state of cells or cell culture conditions could have a determining effect on the dsRNA transcriptome, the underlying dsRNA sensors and the related immune signalling pathways, in essence, calling for further research in the field.‘

Minor points.

>Figure 6C: The Western blot data should be presented as the main figure.

>Done, the order of the figures has been changed.

Not done. I do not see any bands on the Western blot.

We suggest to include the blots in the supplementary information as we feel that a figure including all the individual bands is confusing. Probing of the same blot with multiple antibodies meant that individual membranes were cut into horizontal stripes for the different antibodies resulting in a patchwork of individual snippets. Then, the signals of the dsRNA binding proteins in A375 cells were compared to actin. The same was done with blots from C8161 cells. The relative signals were then compiled and compared in figure 6B. Both the compiled snippets (Supplementary figure 2) as well as the entire blots have been submitted to the ‘Cells’ editorial office.